# An Interpenetrating Alginate/Gelatin Network for Three-Dimensional (3D) Cell Cultures and Organ Bioprinting

**DOI:** 10.3390/molecules25030756

**Published:** 2020-02-10

**Authors:** Qiuhong Chen, Xiaohong Tian, Jun Fan, Hao Tong, Qiang Ao, Xiaohong Wang

**Affiliations:** 1Center of 3D Printing & Organ Manufacturing, School of Fundamental Sciences, China Medical University (CMU), No. 77 Puhe Road, Shenyang North New Area, Shenyang 110122, China; chenqh_xee@163.com (Q.C.); xhtian@cmu.edu.cn (X.T.); jfan@cmu.edu.cn (J.F.); tongh007@163.com (H.T.); aoqiang00@163.com (Q.A.); 2Center of Organ Manufacturing, Department of Mechanical Engineering, Tsinghua University, Beijing 100084, China

**Keywords:** interpenetrating polymer network (IPN), alginate/gelatin hydrogels, three-dimensional (3D) bioprinting, in vitro cell cultures, SH-SY5Y cells

## Abstract

Crosslinking is an effective way to improve the physiochemical and biochemical properties of hydrogels. In this study, we describe an interpenetrating polymer network (IPN) of alginate/gelatin hydrogels (i.e., A-G-IPN) in which cells can be encapsulated for in vitro three-dimensional (3D) cultures and organ bioprinting. A double crosslinking model, i.e., using Ca^2+^ to crosslink alginate molecules and transglutaminase (TG) to crosslink gelatin molecules, is exploited to improve the physiochemical, such as water holding capacity, hardness and structural integrity, and biochemical properties, such as cytocompatibility, of the alginate/gelatin hydrogels. For the sake of convenience, the individual ionic (i.e., only treatment with Ca^2+^) or enzymatic (i.e., only treatment with TG) crosslinked alginate/gelatin hydrogels are referred as alginate-semi-IPN (i.e., A-semi-IPN) or gelatin-semi-IPN (i.e., G-semi-IPN), respectively. Tunable physiochemical and biochemical properties of the hydrogels have been obtained by changing the crosslinking sequences and polymer concentrations. Cytocompatibilities of the obtained hydrogels are evaluated through in vitro 3D cell cultures and bioprinting. The double crosslinked A-G-IPN hydrogel is a promising candidate for a wide range of biomedical applications, including bioartificial organ manufacturing, high-throughput drug screening, and pathological mechanism analyses.

## 1. Introduction

Cells in the body are regulated by a series of internal microenvironments, involving body fluid and extracellular matrices (ECMs). Suitable materials that can imitate the networks of native ECMs allow cells to grow, proliferate, communicate, and transform naturally since their in vivo counterparts, and can be used as ideal matrices for a wide range of biomedical applications [1,2,3].

Hydrogels are 3D hydrophilic polymer networks made of hydrosols containing a large amount of water [4,5,6]. Hydrosols are liquid forms of polymer solutions, which are usually obtained by dissolving polymers in water-based solvents. When hydrosols gel under certain physical (e.g., thermosensitive), chemical (e.g., covalent bonding), or biochemical (e.g., enzymic) conditions, crosslinking among the polymer chains happens, which leads to the formation of hydrogels. Thus, hydrogels are soft and elastic materials, which are generally used above their glass transition temperatures, with outstanding physiochemical and biochemical characteristics similar to soft tissues and organs [7,8,9].

Alginate is a natural anionic polysaccharide derived from brown algae. It is a linear polymer, consisting of 1,4-beta-d-mannuronic acid and alpha-l-guluronic acid, or β-(1→4)-linked D-mannuronic acid (**M**) and α-(1→4)-linked L-guluronic acid (**G**) residues (Figure 1) [10]. A single molecule of alginate can be divided to three parts: “M-blocks, rich in mannuronic acid residues,” “G-blocks, rich in guluronic acid residues,” and “MG-blocks, rich in both.” Alginate molecules can be chemically crosslinked by divalent ions (e.g., Ca^2+^, Sr^2+^, Ba^2+^) by cooperatively binding to guluronic acid residues, so that “G-blocks” can spatially connect each other to generate 3D networks [11]. Alginate hydrogels obtained through divalent ionic crosslinking have been studied for several decades with many products thriving in clinical research and treatment [10,11].

Gelatin, the hydrolysate of collagen, is another natural linear polymer consisting of peptide segments. It has been traditionally used as supporting materials for foodstuffs, drug emulsifiers, and printing impressions [12]. Recently, gelatin has been frequently utilized as ‘bio inks’ with its outstanding thermosensitive and biocompatible properties.

The combination of alginate and gelatin in a hydrogel can maximumly simulate the native extracellular matrix (ECM) components (i.e., proteoglycans) and architectures for broad biomedical applications, such as in vitro 3D cell cultures, tablet drug delivery carriers, and hemostasis dresses [13]. Over the last decade, alginate/gelatin hydrogels have become more popular for complex organ 3D bioprinting with multiple hierarchical vascular and nerve networks. These hydrogels are inborn with unique and prominent biological and physiological properties, such as cell-friendly for encapsulation, mechanical pliable as soft organs, and ease handling for biomedical applications [4,5,6]. However, the mechanical properties of these hydrogels are often not satisfactory for certain biomedical applications where tunable and anti-suture mechanical strengths are required.

In our previous studies, we have created a series of extrusion-based 3D bioprinting technologies and used alginate/gelatin-based hydrogels for a broad range of biomedical applications, such as bio-artificial organ manufacturing, high throughput drug screening, and disease model establishment [14,15,16,17,18,19,20,21,22,23]. Several pioneering crosslinking protocols, such as using glutaraldehyde to crosslink gelatin molecules and Ca^2+^ to crosslink alginate molecules, have been exploited to create interpenetrating networks with stabilized cell-laden 3D constructs. With these protocols we have solved all the bottle neck problems, encountered by tissue engineers [24,25], material (including biomaterial) researchers [26,27,28], stem cell induction experts [29,30,31], pharmaceutists [32,33,34], and tissue/organ cryopreservation scientists [35,36], for more than seven or more decades. In the present study, we use another crosslinking protocol, i.e., both transglutaminase (TG) and Ca^2+^, to prepare alginate/gelatin hydrogels with improved interpenetrating networks for in vitro 3D cell cultures and organ bioprinting.

## 2. Results

### 2.1. Ionic and Covalent Crosslinking Mechanisms of the Interpenetrating Polymer Networks (IPNs)

In Table 1, the gelatin concentration is fixed (i.e., 2.0% *w*/*v*) from Group 1 to 7, during the alginate concentration, which changes from 0.5% to 2.5% (*w*/*v*). From Group 8 to 12, the alginate concentration is fixed while the gelatin concentration changes from 1.5% to 5.0% (*w*/*v*). SH-SY5Y cells are mixed in the hydrosols before TG and calcium chloride solutions are added. The alginate/gelatin hydrogels obtained through covalent and ionic double crosslinking were referred to as alginate-gelatin-interpenetrating polymer networks (A-G-IPNs). While those alginate/gelatin hydrogels obtained through single TG covalent or Ca^2+^ ionic crosslinking were referred to as gelatin-semi-interpenetrating polymer networks (G-semi-IPNs) or alginate-semi-interpenetrating polymer networks (A-semi-IPNs), respectively.

Generally, the single TG crosslinked G-semi-IPN hydrogels are too fragile to be used for further WHC and hardness measurements. The Ca^2+^ ionic crosslinked A-semi-IPN is endurable for WHC and hardness measurements but not for cell-laden biological tests. The A-G-IPN hydrogels are strong enough to be used for further experiments. The structural integrity sequence of the alginate/gelatin hydrogels is A-G-IPN > A-semi-IPN > G-semi-IPN.

The ionic and covalent crosslinking mechanisms of the A-semi-IPN, G-semi-IPN, and A-G-IPN hydrogels are shown in Figure 2. In the A-semi-IPN and A-G-IPN hydrogels, G-blocks in the alginate molecules are chelated by Ca^2+^. In the G-semi-IPN and A-G-IPN hydrogels, gelatin molecules are crosslinked through TG catalyzed covalent linkages. While in the A-G-IPN hydrogels, both TG covalent and Ca^2+^ ionic crosslinks take place.

It is found that the covalent TG crosslinking of gelatin molecules should be arranged before the ionic Ca^2+^ crosslinking of alginate molecules. When the alginate/gelatin hydrosols are first treated with TG, a homogeneously loose and unstable G-semi-IPN hydrogel is obtained. After the G-semi-IPN hydrogel is further treated with Ca^2+^, a solid and stable A-G-IPN hydrogel is produced. The A-G-IPNs in the algainte/gelatin hydrogels demonstrate super structural stabilities compared with the G-semi-IPNs and A-semi-IPNs.

### 2.2. Morphologies of the A-G-IPN Hydrogels

Microstructures of the A-semi-IPN, G-semi-IPN, and A-G-IPN hydrogels are shown in Figure 3. The average pore size sequence of the alginate/gelatin hydrogels is A-semi-IP > A-G-IPN > G-semi-IPN. Especially, the pore shapes in the A-semi-IPN are large and irregular with thick walls and inhomogeneous slices. The pores in the G-semi-IPN hydrogels are small and regular with homogeneous thin walls. The pore morphologies of the A-G-IPN hydrogels are between the A-semi-IPN and G-semi-IPN hydrogels with a middle pore size and wall thickness. This occurs when the alginate/gelatin solution is first treated with TG, a homogeneous inner architecture with small micropores is obtained. Further treated with Ca^2+^ can only make the crosslinks stronger. In contrast, when the alginate/gelatin solution is first treated with Ca^2+^, an irregular internal structure with a large pore size is obtained. This irregular internal structure cannot be changed by further treatment of TG. As a result, the A-semi-IPN in the alginate/gelatin hydrogels is relatively loose and coarse while the G-semi-IPN in the alginate/gelatin hydrogels is relatively dense and exquisite. The covalent TG crosslinking should be arranged before the Ca^2+^ ionic crosslinking if a uniform alginate/gelatin hydrogel is expected.

### 2.3. Water Holding Capacities (WHCs) of the A-G-IPN Hydrogels

Water holding capacity (WHC) refers to the amount of water retained per unit of dry matter. It is an important index of the physiochemical properties of hydrogels. In the present study, the A-semi-IPN and A-G-IPN hydrogels all have excellent water holding capacities, abut 10–30 times as high as their own dry weights. The Ca^2+^ ionic crosslinked hydrogels hold sTable 3D constructs and measurable WHCs. There are no WHC value for the TG covalent crosslinked G-semi-IPN hydrogels, due to the poor structural integrities. The cell-laden G-semi-IPN hydrogels break immediately when they are put into culture medium.

Particularly, WHCs of the A-semi-IPN and A-G-IPN hydrogels are similar (Figure 4). The two curves in Figure 4a,b indicate that the WHCs of the A-semi-IPN and A-G-IPN hydrogels decline in parallel with the increase of the alginate and gelatin concentrations. In Figure 4a, when the gelatin concentration is fixed, the WHC values of the A-semi-IPN, and A-G-IPN hydrogels decrease along with the increase of the alginate concentrations. When the alginate concentrations are between 0.6% to 2.5% (*w*/*v*), the WHCs of the A-semi-IPN and A-G-IPN hydrogels are nearly the same. There are large overlaps in the shadows of the two graphs, which indicates that the WHCs of the A-semi-IPN and A-G-IPN are mainly determined by the Ca^2+^ ionic crosslinking degrees.

In Figure 4b, when the alginate concentration is fixed, the WHC values of the A-semi-IPN, and A-G-IPN hydrogels decrease along while the gelatin concentrations increase. When the gelatin concentrations are between 2.0% to 5% (*w*/*v*), the WHCs of the A-semi-IPN and A-G-IPN hydrogels are closely related. When the gelatin concentration attains a certain degree, i.e., 5% (*w*/*v*), a further increase of the polymer concentration leads to the breakage of the hydrogels without WHCs. This is due to the further increase of the polymer concentration leading to the micropores, where, once full of water molecules, they are occupied by the polymer molecules. The internal microstructures become more compacted. When the micropores in the hydrogels are crowded by the additional polymer molecules, the WHCs drop sharply until the constructs are broken. Different crosslinking mechanisms and sequences have brought about different polymer chain bonding tightness values (i.e., mechanical strengths), spatial microstructures, and WHCs, which are consistent with the results of SEM images in Figure 3.

### 2.4. Hardness of the A-G-IPN Hydrogels

Hardness is the resistance of a solid material against various permanent shape changes when a compressive force is applied. The formation of the IPN structures can significantly enhance the hardness of the alginate/gelatin hydrogels. Due to the poor structural integrities of the TG covalent crosslinked gelatin molecules, the hardness of the G-semi-IPN hydrogels is unmeasurable. The hardness of the A-semi-IPN and A-G-IPN hydrogels are reasonable.

As shown in Figure 5, the hardness of the A-G-IPN hydrogel is clearly higher than that of the A-semi-IPN hydrogel when the alginate concentrations are between 0.75% to 1.25% (*w*/*v*). Due to a further increase of the alginate concentrations from 1.5% to 2.5% (*w*/*v*), the values of the two groups are close to each other. When the concentration of the alginate is 2.5% (*w*/*v*). Both hardness values of the A-semi-IPN and A-G-IPN hydrogels attain the highest. The low proportion of alginate (i.e., alginate concentration between 0.75% to 1.2%) makes the TG covalent crosslinking an outstanding role for the hardness achievement of the A-G-IPN hydrogels (Figure 5a). However, when the alginate concentration is increased to 1.25% (*w*/*v*), the TG covalent crosslinking role is reduced sharply along with the increase of the alginate concentrations. These changes can be distinguished from the coupled columns in Figure 5a. The high proportion of alginate makes the TG’s role overwhelmingly or completely covered by the Ca^2+^ ionic crosslinks.

In contrast, the hardness of the A-G-IPN hydrogels is much higher than that of the A-semi-IPN hydrogels when the alginate concentration is fixed (i.e., 0.75% *w*/*v*). As shown in Figure 5b, the low proportion of gelatin, i.e., 1.5% (*w*/*v*), makes a relative high hardness of the hydrogels. In the low gelatin concentration samples, the TG crosslinking seems sufficient for the hardness achievement. When the concentration of gelatin is risen to 2.5% (*w*/*v*), the hardness values of the A-semi-IPN and A-G-IPN hydrogels decline significantly. In these cases, the covalent crosslinking of the gelatin molecules is less helpful for the hardness achievement compared with the Ca^2+^ ionic crosslinked alginate molecules. These results indicate that the Ca^2+^ ionic crosslinks have played a major role in the maintenance of the structural integrity of the A-semi-IPN and A-G-IPN hydrogels. Different crosslinking values contribute differently to the hardness of the composite alginate/gelatin hydrogels. Only within certain polymer concentrations, the hardness measurement makes sense and the results are meaningful.

### 2.5. Cell States in the A-G-IPN Hydrogels

Cell states in the A-G-IPN hydrogels were characterized through an optical microscope, scanning electron microscopy (SEM), and a laser confocal microscope (LSM). Based on the optimization results of the polymer concentrations, a combination of 0.75% (*w*/*v*) of alginate and 2% (*w*/*v*) of gelatin was used for in vitro cell cultures. Figure 6 shows that the states of the human neuroblastoma (SH-SY5Y) cells in the A-G-IPN hydrogels look like little balls during the 7 days of in vitro cultures. After 4 and 7 days of in vitro cultures, a lot of cell clusters appear (Figure 6c,d), which indicates that cell divisions are very active in the A-G-IPN hydrogels. In contrast, SH-SY5Y cells cultured on the 2D plastics behave like flat spindles (or shuttles) adhered to the plate (Figure 6a).

Figure 7 shows that, after freeze-drying, most of the cells are encapsulated in the A-G-IPN hydrogels. Cell connections can be found in the crevices of the dried alginate/gelatin hydrogels. Compared with those of the day 7 sample (Figure 7a), there are more cell clusters on the day 14 sample (Figure 7c).

In Figure 8, there is no stained red fluorescence, and the area of green fluorescence extends gradually with the increase of culture time. This means that all the SH-SY5Y cells are alive in a green color with AO/PI staining after 1, 3, and 5 days of in vitro cultures. Few cell aggregates can be found in the sample during day 1 (Figure 8a,d). Most of the cells are in round shapes and scattered in the A-G-IPN. The gaps among the separated cells are large. On the contrary, the number and size of the cell aggregates on day 3 are clearly higher and larger compared with those on day 1 (Figure 8b,e). Distinguishingly, very large cell aggregates accompanying a lot of vortex-like structures are prominent in the samples of day 5 (Figure 8c,f). Cell-cell and cell-matrix communications may happen through the vortex-like structures.

### 2.6. Cell Proliferation Rate in the A-G-IPN Hydrogels

Cell proliferation rate in the A-G-IPN hydrogel is characterized using a CCK-8 kit on day 1, 2, 3, 5, and 7. As shown in Figure 9, the cell viability increases constantly without a plateau during the 7 days of in vitro cultures. This result is consistent with the confocal images shown in Figure 8, where SH-SY5Y cells are all in living states (green) with augmented aggregates over time.

### 2.7. Histological Analysis

Due to the poor mechanical properties of the A-semi-IPN and G-semi-IPN hydrogels, the cell-laden 3D constructs are broken during the first several days of in vitro cultures. As such, there are no historical images of the cells in these constructs.

In Figure 10, SH-SY5Y cells are stained dark red or purple in the A-G-IPN hydrogels. There is a clear boundary between the cytoplasm and nucleus in the stained cells. Cells are in round shapes either after 7 or 14 days of in vitro cultures. After 7 days of an in vitro culture, cell divisions can be distinguished through the connected two or three neighbour cells (Figure 10c). A few small cell aggregates can be visually observed. After 14 days of an in vitro culture, the cell sizes and shapes remain (Figure 10d). However, cell aggregates are clearly much larger. These results are in accordance with the optical microscope (Figure 6), SEM (Figure 7), and acridine orange (AO)/propidium iodide (PI) (i.e., AO/PI) staining laser confocal microscope (LSM) (Figure 8) results. Cells have enough spaces to grow, proliferate, and spread in the A-G-IPN hydrogels. The double crosslinked alginate/gelatin interpenetrating network is stable (or strong) enough to provide the encapsulated cells with necessary gas (such as oxygen), water, and nutrients for more than two weeks.

### 2.8. Three-Dimensional (3D) Organ Bioprinting

Conventional in vitro cell culture strategies have many limitations for complex organ manufacturing. These limitations involve lacking in large scale-up, high-throughput, and structural replication of the products. Over the last decade, we have created a series of extrusion-based 3D bioprinting technologies to manufacture complex organs in a layer-by-layer manner under the precise control of computer-aided design (CAD) models [14,15,16,17,18,19,20,21,22,23]. Most of these extrusion-based 3D bioprinting technologies employ spiral-squeezing presses to deposit cell-laden hydrogels mimicking natural organs. In the present study, a new 3D bioprinter, with three spiral-squeezing press nozzles, is first tested for organ 3D bioprinting.

As shown in Figure 11, 3D cell-laden constructs with high structural fidelity of the extruded filaments are obtained using the predesigned alginate/gelatin and gelatin hydrogels and optimized crosslinking methods. The un-crosslinked gelatin filaments, endowed as a sacrificial material, are removed immediately following 3D bioprinting, leaving behind more go-through channels in the 3D constructs consisting of cell-laden alginate/gelatin hydrogels. The resolution is as high as 10 micron, which is similar to former studies [14,15,16,17,18,19,20,21,22,23]. Additionally, it is realized that the printing resolution is not the main principal contradiction in 3D organ bioprinting. For most of the living cells, their sizes are between 10–20 μm. 3D bioprinting makes sense only when the thickness of the printing filaments is larger than 10 μm. Within this thickness, cells can adjust themselves, according to the microenvironments. Before bioprinting, the gelatin molecules in the alginate/gelatin hydrogels are partially crosslinked using TG. The weak TG crosslinked gelatin molecules have enhanced the viscosity of the alginate/gelatin hydrogels, but not the printing resolution. Compared with the Ca^2+^ ionic crosslinked alginate molecules, the TG covalent crosslinked gelatin molecules act more like a dispersal medium, which enables subtle control of the micro pore size and distribution by adjusting the volume ratio of the alginate/gelatin solutions. In general, the double crosslinked A-G-IPN hydrogels can serve as stable templates for cell encapsulation and 3D bioprinting. As a new strategy to engineer porous cell-laden 3D constructs, this new double crosslinking protocol holds the capacity to be widely used in future complex organ-manufacturing areas.

## 3. Discussion

An IPN of alginate/gelatin hydrogel, crosslinked through both TG covalent and Ca^2+^ ionic crosslinking, is presented in this section. The crosslinked alginate/gelatin IPNs in the hydrogels can mimic the natural ECMs in human soft organs with suitable mechanical, hydration, and biological properties for 3D in vitro cultures and bioprinting [14,15,16,17,18,19,20,21,22,23]. It is found that changes in the polymer concentrations and crosslinking sequence directly affect the microstructures, hardness, and WHCs of the alginate/gelatin hydrogels. TG covalent crosslinking should be arranged before Ca^2+^ ionic crosslinking to obtain a homogeneous internal microstructure. The structural integrity sequence of the hydrogels is A-G-IPN > A-semi-IPN > G-semi-IPN. SH-SY5Y cells can grow well in the A-G-IPN hydrogels for more than two weeks.

The mechanism of alginate ionic crosslinking through divalent ions is that the neighbour G-blocks in alginate molecules form “egg-box”-like chelate structures with divalent ions (Figure 2a) [37,38,39]. When calcium chloride solution is added into the alginate-containing solutions, Ca^2+^ ions interact with the alginate molecules through coordination bonds and polyelectrolyte effects. The coordination bonds possess strong polarities and contribute to the heterogeneous hydrogel formation. After ionic crosslinking, the alginate-containing hydrosols become hydrogels with significantly improved mechanical strengths.

Thus, the Ca^2+^ crosslinked alginate containing hydrogels can be regarded as a polyelectrolyte gel consisting of an IPN crosslinked with ionizable groups in a liquid phase. This is a reversible chemical crosslinking among the G-blocks of the intertwined alginate chains. When the Ca^2+^ crosslinked structures are put into a cell culture medium with a low concentration of Ca^2+^, the chelated Ca^2+^ ions can dissolve into the liquid phase in several days, which leads to the breakage of the 3D constructs [40,41,42,43,44]. Further solidification of the 3D constructs is necessary for long-term in vitro cell cultures and 3D organ products.

The mechanism of gelatin crosslinking is that the glutamine and lysine residues on the peptide segments of the gelatin chains can be catalyzed by enzyme TG with covalent and hydrogen bonds, which results in the sol-gel (i.e., glass) transition of gelatin solution at 28 °C (Figure 2b) [12,16]. After the enzyme-catalyzed reaction, the covalent bonds possess weak polarities, which contribute to the homogeneous hydrogel formation. The peptide segments tend to transform into multiple-branching constructions. When a balance between the polarizing and depolarizing factors attains a gelatin-containing hydrogel with weak mechanical strength can be obtained. The covalent and hydrogen bonds have played a less prominent role in the structural integrity maintenance of the A-semi-IPN and A-G-IPN hydrogels compared with the Ca^2+^ ionic crosslinks.

The A-semi-IPN hydrogel is formed through crosslinking alginate molecules in the alginate/gelatin hydrosol. Similarly, the G-semi-IPN hydrogel is formed by crosslinking gelatin molecules in the alginate/gelatin hydrosol. Meanwhile, the A-G-IPN hydrogel is formed by crosslinking both alginate and gelatin molecules in the alginate/gelatin hydrosol. In the A-semi-IPN, G-semi-IPN, and A-G-IPN hydrogels, water molecules can occupy the micropore spaces and form hydrogen bonds with the hydrophilic groups in the polymer chains. When the alginate/gelatin solutions are crosslinked by Ca^2+^ or TG, A-semi-IPN or G-semi-IPN are obtained. Some of the un-crosslinked polymers may dissolve in culture medium during the later in vitro cultures. These polymers can be regarded as porogenic agents, which benefit micropore enlargement in the hydrogels. For example, within certain alginate/gelatin concentrations, WHCs of the hydrogels decrease with the addition of the component polymers. When a large amount of gelatin molecules is added, the gaps between neighbour crosslinked alginate chains can be enlarged. The un-crosslinked gelatin molecules can be regarded as porogenic agents when they dissolve into culture medium during the later in vitro 3D cultures [45,46,47,48,49,50]. When the un-crosslinked alginate molecules dissolve out, they can also be regarded as porogenic agents. Compared with the double crosslinked A-G-IPN hydrogels, the A-semi-IPN and G-semi-IPN all have some clear shortcomings for the long-term structural integrity maintenance during the in vitro 3D cell cultures and bioprinting processes (Figure 3, Figure 4, Figure 5, Figure 6, Figure 7, Figure 8, Figure 9, Figure 10 and Figure 11).

The combination of the algae-derived polysaccharide alginate and animal protein-derived gelatin has better mimicked the components of ECMs around human cells in soft organs [51,52,53,54]. Suitable A-G-IPNs with optimized polymer concentrations have been achieved through different alginate and gelatin compositions and crosslinking sequences. The similar water-holding capacities of the A-semi-IPN and A-G-IPN hydrogels indicate that the ionic crosslinked alginate molecules have played a major role in the stabilization of the IPNs. This can be certified by the hardness test results. The enzymatic TG crosslinking has played a less important role in the structural integrity, WHC, and hardness achievements, but can change the internal microstructures of the hydrogels. The improved physiochemical properties of the alginate/gelatin hydrogels have been confirmed by the in vitro 3D cell culture and bioprinting results (Figure 6, Figure 7, Figure 8, Figure 9 and Figure 10) [14,15,16,17,18,19,20,21,22,23].

There are great differences for SH-SY5Y cells cultured on the 2D plates and in the 3D hydrogels. SH-SY5Y cells cultured on the 2D plastics are flat shuttle-like with long pseudopods pseudopodia due to the intrinsic adherent properties (Figure 6a). Cells in the 3D hydrogels exhibit a spherical morphology because the IPNs can provide cells with enough mechanical support and limited space to spread. The 3D IPNs surrounding the cells are similar to the native cell survival microenvironments with enough binding sites. The cell shapes can be defined by the nearest ionic crosslinked alginate/gelatin matrix, which is clearly controlled by the ingredient polymer concentrations and pertinent crosslinking degrees. The A-semi-IPNs contribute substantially to the long-term 3D structural integrities and cell metabolic activities.

It is assumed that, with the proliferation of cells, the size of the micropores in the A-semi-IPN and A-G-IPN hydrogels can be enlarged. This may be due to the following reasons: (1) the un-crosslinked gelatin molecules may dissolve out over time, and (2) some of the ionic or covalent crosslinks may break down in the culture medium. Cells can penetrate the IPNs through the micropores. The A-semi-IPNs and A-G-IPNs can provide cells with expansive space to proliferate before they are completely broken down [28,29,55,56,57,58]. These can be deduced from the constant augmented cell aggregates in the long-term in in vitro cell cultures and bioprinting constructs (Figure 6, Figure 8 and Figure 11).

3D bioprinting is a new biomaterial shaping technology developed in the early 2000s. It is based on digital model files to construct solid cell-laden objects using connectable biomaterials. Over the past decade, extrusion-based 3D bioprinting technologies have demonstrated outstanding advantages in biomedical fields, especially in bioartificial organ manufacturing areas [59,60]. The 3D printed bioartificial organs hold the highest potentials to temporarily or permanently repair, replace, or restore their defective/failure counterparts [61,62,63].

Primary experiments show that SH-SY5Y cells grow very well in the grid 3D constructs during the five days of in vitro cultures. After 3D bioprinting and crosslinking, the IPN structures in the cell-laden gelatin/alginate hydrogels can be maintained while the biocompatibility of the gelatin/alginate hydrogel can be retained. The double crosslinked INPs have shown their potential as a viable ‘bio ink’ with tunable physicochemical and biochemical properties, similar to native cell growth 3D environments. The combination of TG covalent and Ca^2+^ ionic crosslinking is an effective way to improve the structural integrity of the 3D constructs, functional preservation of the natural ECM-like polymers, and internal structures of the simulated 3D environments.

## 4. Materials and Methods

### 4.1. Materials

Sodium alginate (Ref-W201502, Mη 222, G/M roughly 1.13, viscosity 5–40 cps at 1% and 25 °C) and gelatin (from porcine skin, type B) were purchased from Sigma (Sigma, St. Louis, MO, USA). Transglutaminase (TG) (100 U/g, Dongsheng, Shanghai, China) was purchased from Dongsheng Co., Ltd., Shanghai, China). Other reagents used in the experiments were all of an analytical grade purchased from Beyotime Biotechnology (Shanghai, China).

### 4.2. Preparation of Hydrogels

Powder sodium alginate and gelatin were weighted and put into a phosphate buffer saline (PBS) buffer according to the ratio of weight/volume (*w*/*v*) in Table 1. After the powders were dissolved in a water bath at 70 °C, a well-mixed solution was obtained. When the solution was cooled down, a 10% (*w*/*v*) TG solution was added for covalent crosslinking gelatin molecules 6 h before a 2% (*w*/*v*) calcium chloride (i.e., CaCl_2_, Sinopharm Chemical Regent Beijing Co., Ltd.) solution was added for ionic crosslinking alginate molecules for 2 min [37]. Additional amounts of TG and CaCl_2_ solutions were applied when the polymer concentrations were increased, fully ensuring reactions of the crosslinking.

### 4.3. Water Holding Capacity Test

Water holding capacities (WHCs) of the A-G-IPN, A-semi-IPN, and G-semi-IPN samples, prepared according to Table 1, were tested using a modified method as reported [38]. First, wet weight (W_w_) was measured after the supernatant liquid over samples was removed. Then the samples were freeze-dried to measure dry weight (W_d_). One ml of the hydrosols was used for each of the samples. The water holding capacity (g/g sample) was calculated using the following formula.
WHC = (W_w_ − W_d_)/W_d_(1)

### 4.4. Hardness Test

A self-made cylinder mold, made of well-permeating filter paper, was used to prepare the hydrogel samples. When the alginate/gelatin containing solutions, prepared according to Table 1, were poured into the mold with a diameter of 10 mm and a height of 20 mm, they were crosslinked with TG and/or calcium chloride (i.e., CaCl_2_) solutions, respectively. The hardness testing mechanism is described in Figure 12. Briefly, samples were placed on the platform of Shore durometer (HT-6510OO, Lantai, Shenzhen, China), and the R1.2 hemispherical probe of the Shore durometer was used to measure the hardness [39]. When the handle was pushed down gently, the number on the counter board changed accordingly. When the hemispherical probe was down to 10 mm, the maximum number appeared on the counter board was recorded. Each group was tested with six duplicated samples.

### 4.5. Scanning Electron Microscopy (SEM) Observation

Some A-semi-IPN, G-semi-IPN, and A-G-IPN hydrogel samples were prepared in 12-well plate according to the *w*/*v* ratio of Group 2 in Table 1, before being frozen in liquid nitrogen and dried in vacuum. The samples were immobilized with a 2.5% glutaraldehyde solution (Scientifc Phygene, China) at 4 °C for 30 min before being bound to a double-sided adhesive tape and sputter-coated with gold in order to make the fracture surfaces conductive. Each sample was tested six times using a scanning electron microscopy (SEM, VEGA3, TESCAN, Czech Republic).

### 4.6. Cell Cultures in the A-G-IPN Hydrogels

SH-SY5Y cells are an adrenergic clone of the human neuroblastoma cell line SK-N-SH. In the present study, human NB SH-SY5Y cells obtained from the American Type Culture Collection (ATCC, Rockville, MD, USA) were used for the experiments. The cells were first cultured on two-dimensional (2D) plastic dishes for proliferation before being retrieved and suspended in the previously mentioned 0.75% (*w*/*v*) alginate/2% (*w*/*v*) gelatin solutions at a density of 2 × 10^7^ cells/mL for in vitro cultures. Then 10% (*w*/*v*) TG solution was used to crosslink the gelatin molecules in the hydrosols at 37 °C for 6 h. Furthermore, 2% (*w*/*v*) calcium chloride solution was used to crosslink the alginate molecules in the samples for 2 min. The samples were vibrated during the crosslinking processes to make cells suspend evenly in the hydrosols and the crosslinking agents cover the hydrosols. One ml of alginate/gelatin solution was used in each well of a 12-well plate.

The cell-laden samples were subsequently cultured in Dulbecco’s modified Eagle’s medium (DMEM, Gibco, Grand Island, NY, USA) containing 10% heat-inactivated fetal bovine serum (FBS, Gibco, Grand Island, NY, USA) and 1% penicillin/streptomycin (Gibco, Grand Island, NY, USA) at 37 °C, 5% CO_2_ for more than 8 days. Media was changed every two days.

The optical microscope was used frequently to observe the cell states in the transparent A-G-IPN hydrogels at different in vitro culture periods. SH-SY5Y cells cultured on a 2D plastic were used as a control. SEM analysis on cross sections of the samples was carried out after one week and two weeks of in vitro cultures as mentioned above.

### 4.7. Cell Viability

Cell survival status were assessed by an acridine orange (AO)/propidium iodide (PI) double staining kit, i.e., fluorescent live/dead viability assay kit (BestBio, Beijing, China), according to the instruction. The samples were immersed in a phosphate buffer saline (PBS) containing 5 μL of AO and 10 μL of PI and incubated in dark at 4 °C for 10 min. After being washed with PBS three times, they were checked using a laser confocal microscope (LSM, N1R, Nikon, Japan) at 488 nm exciting light. Dead cells were stained a red color while living cells were stained a green color. This procedure was repeated every other day for 5 days.

The cell viability assay was performed using a CCK-8 kit (Vazyme, Nanjing, China) following the instructions. SH-SY5Y cells were encapsulated in the 0.75% (*w*/*v*) alginate/2% (*w*/*v*) gelatin hydrosols with a density of 2 × 10^6^ cells/mL. After the cell-laden hydrosols were poured in the wells of a 96-well plate. Iron and covalent crosslinking took place before culture medium was added for in vitro cultures. A total of 100 uL hydrosol was used for each well. Cell-free hydrogels with similar volumes were used as controls. The culture medium was changed every day. After a certain period, the constructs were washed with PBS. The detection reagent (100 μL medium + 20 μL CCK-8) was added to each well in order to complete staining, which takes 2 h [29]. After staining, the solution in each well was transferred to a blank 96-well plate to detect the optical density (OD) at 450 nm exciting light (Thermo Fisher Scientific, Waltham, MA, USA). The mean OD values of the A-G-IPN groups were expressed as OD_t_, while the control groups were expressed as OD_n_. Cell viability (CV) was calculated via the following formula.
(2)$$\mathrm{CV}=\frac{{\mathrm{OD}}_{t}-{\mathrm{OD}}_{n}}{{\mathrm{OD}}_{n}}$$

Each experiment was performed in three replicas.

### 4.8. Histological Observation

Due to the poor structural integrities of the A-semi-IPN and G-semi-IPN hydrogels, the cell-laden 3D constructs were broken during the first several days of in vitro cultures. After one week and two weeks of in vitro cultures, cells in the A-G-IPN hydrogels were immobilized with 4% fixative solutions (Solarbio, Beijing, China) for 12 h at 4 °C, and then immersed in 20%, 30% (*w*/*v*) sucrose solutions until the constructs sunk to the bottom. The constructs were embedded in an optimum cutting temperature (O.C.T.) compound (Solarbio, Beijing, China) and cut with a freezing microtome (Leica, Germany) to obtain 8-μm thick sections. After air seasoning, the sections were stained with haematoxylin and eosin (HE).

### 4.9. Three-Dimensional (3D) Bioprinting

A cell-laden 0.75% (*w*/*v*) alginate/2% (*w*/*v*) gelatin hydrogel was prepared according to former experiments [14,15,16,17,18,19,20,21,22,23]. A cell-free gelatin solution 20% (*w*/*v*) was prepared especially with the powder dissolving in a 2% (*w*/*v*) CaCl_2_ solution. A circular grid pattern with a thickness of 5–10 mm, a mesh size of 0.2–2.5 mm, and a radius of 10–30 mm was designed using a software package (Microsoft, AT6400) equipped in a home-made three-nozzle 3D bioprinter. Human neuroblastoma SH-SY5Y cells were first mixed into the gelatin/alginate solution at a density of 2 × 10^6^ cells/mL. A 10% (*w*/*v*) TG solution was added for covalent crosslinking gelatin molecules for 6 h at 37 °C. After mixing thoroughly, 10 mL of the mixture was loaded into one of the barrels, and 30 mL of the cell-free gelatin solution was loaded into another barrel at 70 °C, waiting for 30 min at 20 °C before bioprinting. Two nozzles were employed. One nozzle with an inner diameter of 0.16 mm was used to deposit the cell-laden ‘bioink’, while another nozzle with an inner diameter of 0.41 mm was used to deposit the cell-free gelatin hydrosol onto a plastic plate under the control of a computer-aided design (CAD) model. The layer thickness of the z-axis was set at 0.4 mm. The program was run 10 times, consecutively, at the same position to ensure the generation of a 3D configuration. During the 3D printing process, the two nozzles worked alternately. The nozzle for the cell-laden ‘bio ink’ deposition was run two times before the nozzle for the gelatin hydrosol deposition was run one time. After 3D printing, the bottom and top layers of the 3D constructs were all cell-laden alginate/gelatin hydrogels. When the 3D printing process was finished, the 3D constructs were put into a Petri dish containing 2% (*w*/*v*) CaCl_2_ for alginate crosslinking. The pure gelatin filaments were washed for 30 min at 37 °C. Lastly, the 3D constructs were moved into an incubator and cultured with dulbecco’s minimum essential medium (DMEM) containing 10% FBS and 1% penicillin/streptomycin at 37 °C, 5% CO2 for more than five days. The culture medium was changed every two days and the growth states of the cells were monitored daily via phase-contrast microscopy. On day 5, a piece of the construct was stained with an AO/PI kit and checked using the LSM mentioned above.

### 4.10. Statistics Analysis

The results were presented as the mean ± standard deviation (SD) where applicable. Statistical analysis was performed using a Statistical Product and Service Solutions (SPSS) 22.0 software (Chicago, IL, USA) and plotted by a GraphPad Prism 8.0 software (USA). The Student’s t-test was used for a comparison between the A-semi-IPN and A-G-IPN groups under the same polymer concentrations in water holding capacity (WHC) and hardness measurements. The one-way analysis of variance (ANOVA) method was used for comparing the cell viability (CV) values. A P-value less than 0.05 was considered statistically significant.

## 5. Conclusions

An IPN is successfully prepared by sequentially crosslinking gelatin molecules using covalent (i.e., enzymic) TG and alginate molecules using ionic Ca^2+^ in the alginate/galatin hydrosols. The properties of A-semi-IPN, G-semi-IPN, and A-G-IPN in the hydrogels are confirmed through physiochemical and biochemical characterizations, such as optical microscope, SEM, and LSM images, WHCs, and hardness results. The Ca^2+^ ionic crosslinked alginate molecules have played a major role in the maintenances of the 3D structural integrities, and the achievements of the expected physiochemical properties, such as hydration capacities and hardness values (i.e., mechanical strengths). The TG covalent crosslinked gelatin molecules benefit from suitable micropore formation in the alginate/gelatin hydrogels. Optimized internal microstructures, WHCs, and hardness values can be achieved by changing the covalent and ionic crosslinking sequences and polymer concentrations of the alginate/gelatin hydrogels. The ECM mimicked alginate/gelatin IPNs can provide cells with suiTable 3D environments for survival, growth, and proliferation. SH-SY5Y cells grow and proliferate well in the A-G-IPN hydrogels for more than two weeks without breaking down the 3D constructs. These hydrogels are promising candidates for widespread biomedical applications, such as in vitro 3D cell cultures, bioartificial organ manufacturing, high-throughput drug screening, and physiological mechanism analyses.

## Figures and Tables

**Figure 1 molecules-25-00756-f001:**
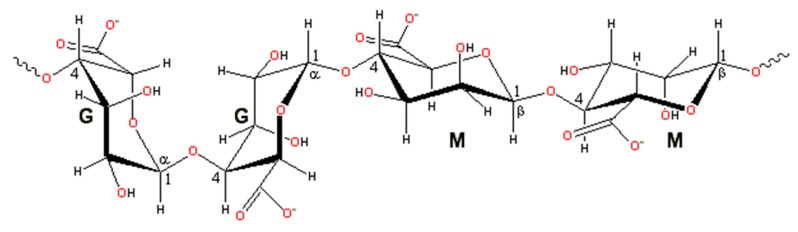
Structure units of alginate molecule (from Wikipedia, the free encyclopedia).

**Figure 2 molecules-25-00756-f002:**
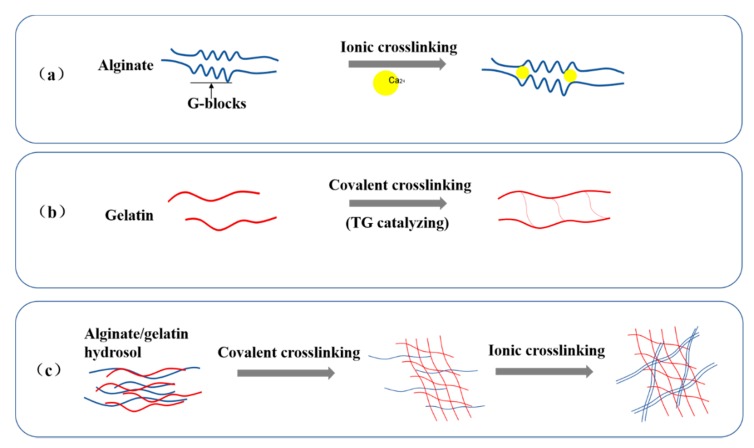
Chemical description of ionic crosslinking of alginate molecules, covalent crosslinking of gelatin molecules, and both covalent and ionic crosslinking of alginate/gelatin molecules. (**a**) G-blocks in two alginate chains are chemically (i.e., ionic) crosslinked by Ca^2+^. (**b**) Transglutaminase (TG) catalyzed covalent linkages between two gelatin molecules. (**c**) An interpenetrating network in an alginate/gelatin hydrogel formed through both TG covalent and Ca^2+^ ionic crosslinks.

**Figure 3 molecules-25-00756-f003:**
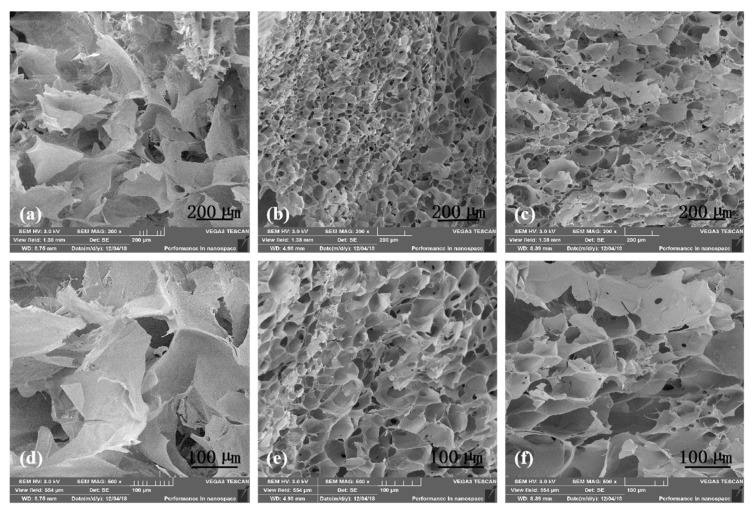
Scanning electron microscopy (SEM) micrographs of the alginate/gelatin hydrogels: (**a**) alginate-semi-interpenetrating polymer network (A-semi-IPN), (**b**) gelatin-semi-interpenetrating polymer network (G-semi-IPN), (**c**) alginate-gelatin-interpenetrating polymer network (A-G-IPN), (**d**) a magnified image of (**a**,**e**) a magnified image of (**b**), and (**f**) a magnified image of (**c**).

**Figure 4 molecules-25-00756-f004:**
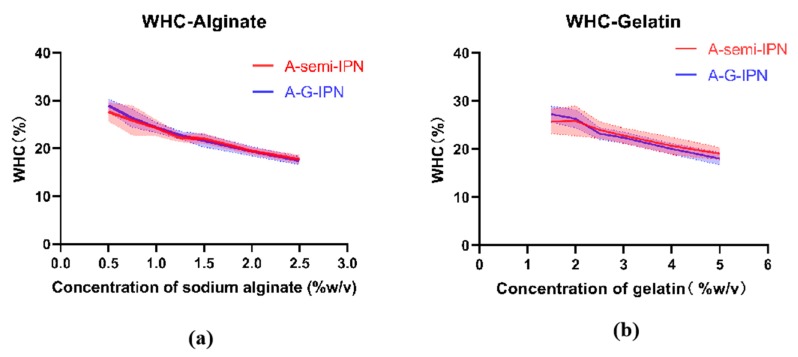
Water holding capacities (WHCs) of the alginate-semi-interpenetrating polymer network (A-semi-IPN) and alginate-gelatin-interpenetrating polymer network (A-G-IPN). (**a**) WHCs of the A-semi-IPN and A-G-IPN hydrogels along with different alginate concentrations. (**b**) WHCs of the A-semi-IPN and A-G-IPN hydrogels along with different gelatin concentrations. There are no statistical significance between the A-semi-IPN and A-G-IPN hydrogels in WHCs.

**Figure 5 molecules-25-00756-f005:**
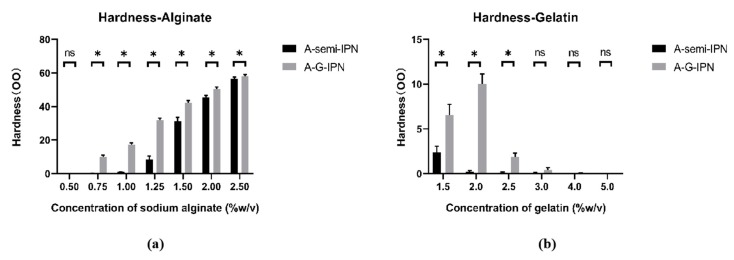
Hardness of the alginate-semi-interpenetrating polymer network (A-semi-IPN), and alginate-gelatin-interpenetrating polymer network (A-G-IPN): (**a**) hardness of the A-semi-IPN and A-G-IPN with different alginate concentrations. (**b**) Hardness of the A-semi-IPN and A-G-IPN hydrogels with different gelatin concentrations. * means that there are statistical significances between the A-semi-IPN and A-G-IPN groups (*p* < 0.05).

**Figure 6 molecules-25-00756-f006:**
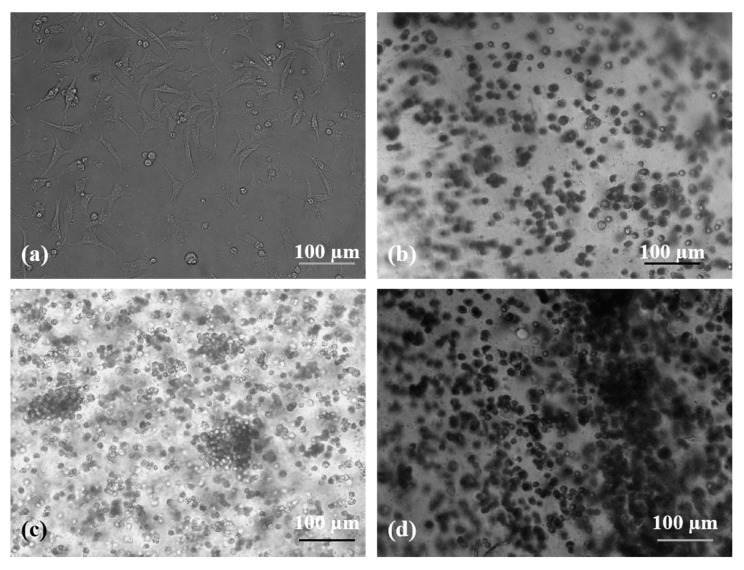
Human neuroblastoma SH-SY5Y cells cultured on a 2D plastic and in a 3D alginate-gelatin-interpenetrating polymer network (A-G-IPN): (**a**) an optical microscope image of SH-SY5Y cells cultured on a 2D plastic for 1 day. (**b**) An optical microscope image of SH-SY5Y cells cultured in a 3D A-G-IPN hydrogel for 1 day. **(c)** An optical microscope image of SH-SY5Y cells cultured in a 3D A-G-IPN hydrogel for 4 days. (**d**) An optical microscope image of SH-SY5Y cells cultured in a 3D A-G-IPN hydrogel for 7 days.

**Figure 7 molecules-25-00756-f007:**
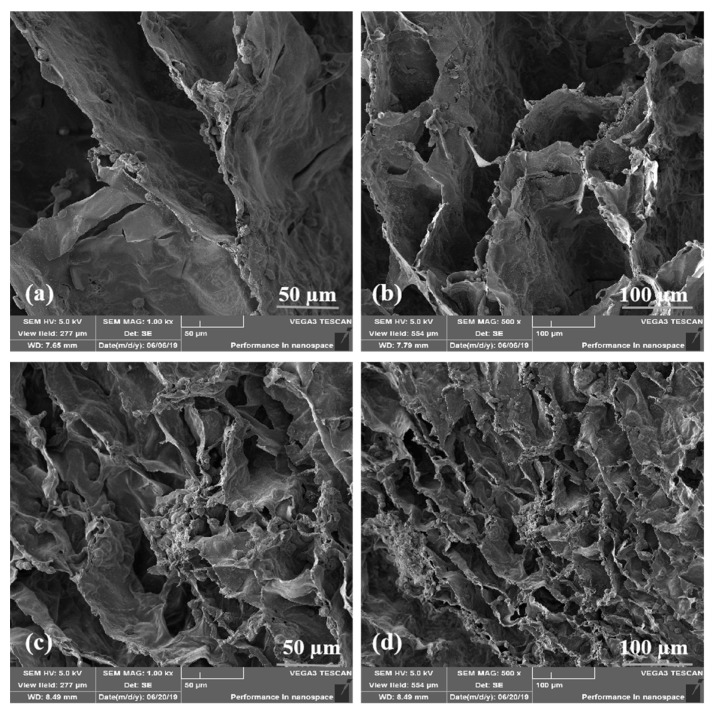
Human neuroblastoma SH-SY5Y cells encapsulated in the alginate-gelatin-interpenetrating polymer network (A-G-IPN) after freeze-drying. (**a**) A scanning electron microscopy (SEM) image showing SH-SY5Y cells encapsulated in the A-G-IPN hydrogel after 7 days of an in vitro culture. (**b**) A magnified image of (**a**). (**c**) A SEM image showing SH-SY5Y cells encapsulated in the A-G-IPN hydrogel after 14 days of an in vitro culture. (**d**) A magnified image of (**c**).

**Figure 8 molecules-25-00756-f008:**
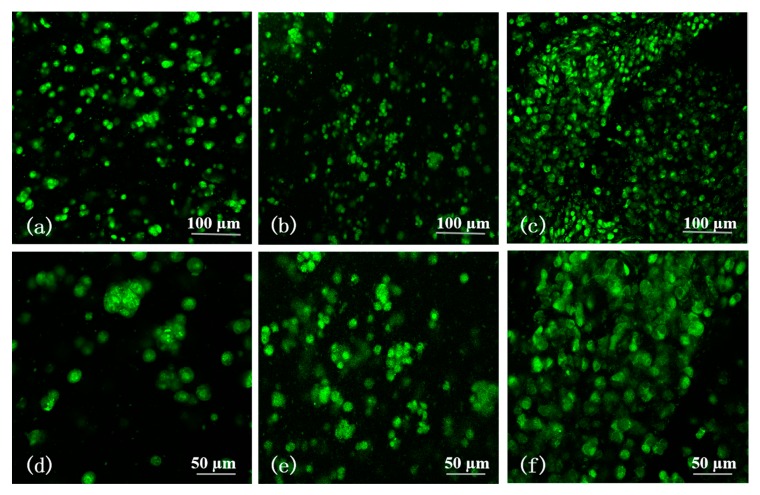
Viability of human neuroblastoma SH-SY5Y cells encapsulated in the alginate-gelatin-interpenetrating polymer network (A-G-IPN) with acridine orange (AO)/propidium iodide (PI), (i.e., AO/PI), staining: (**a**) a laser confocal microscope (LSM) image showing that all the SH-SY5Y cells are alive in a green color with AO/PI staining after 1 day of an in vitro culture. (**b**) A LSM image showing that all the SH-SY5Y cells are alive in a green color with AO/PI staining after 3 days of an in vitro culture. (**c**) A LSM image showing that all the SH-SY5Y cells are alive in a green color with AO/PI staining after 5 days of an in vitro culture. (**d**) A magnified image of (**a**). (**e**) A magnified image of (**b**). (**f**) A magnified image of (**c**).

**Figure 9 molecules-25-00756-f009:**
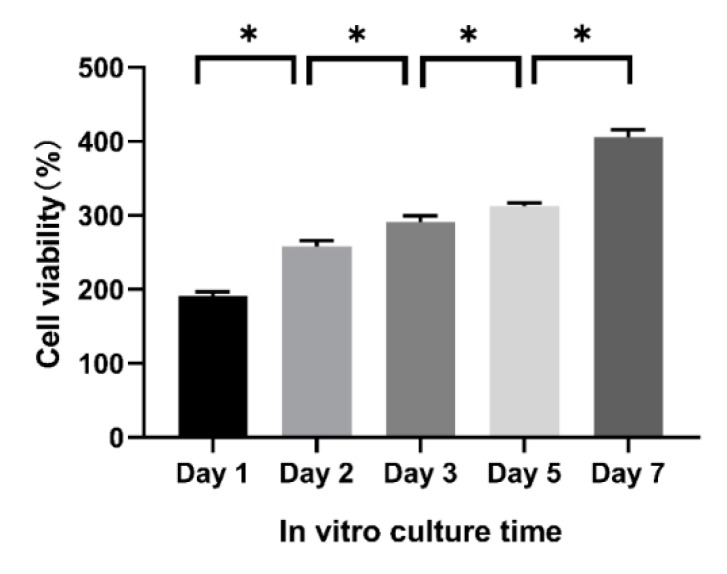
Human neuroblastoma SH-SY5Y cell proliferation rate in the alginate-gelatin-interpenetrating polymer network (A-G-IPN). * means that there are statistical significances between the two adjacent culture periods (*p* < 0.05).

**Figure 10 molecules-25-00756-f010:**
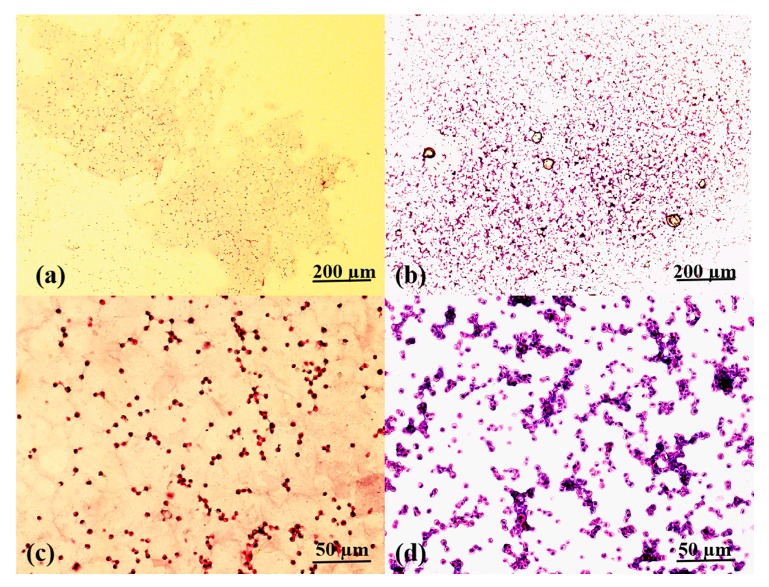
Haematoxylin and eosin (HE) staining of the cell-laden alginate-gelatin-interpenetrating polymer network (A-G-IPN): (**a**) a HE image of SH-SY5Y cell-loaded A-G-IPN hydrogel after 7 days of an in vitro culture. (**b**) A HE image of SH-SY5Y cell-loaded A-G-IPN hydrogel after 14 days of an in vitro culture. (**c**) A magnified image of (**a**), (**d**) A magnified image of (**b**).

**Figure 11 molecules-25-00756-f011:**
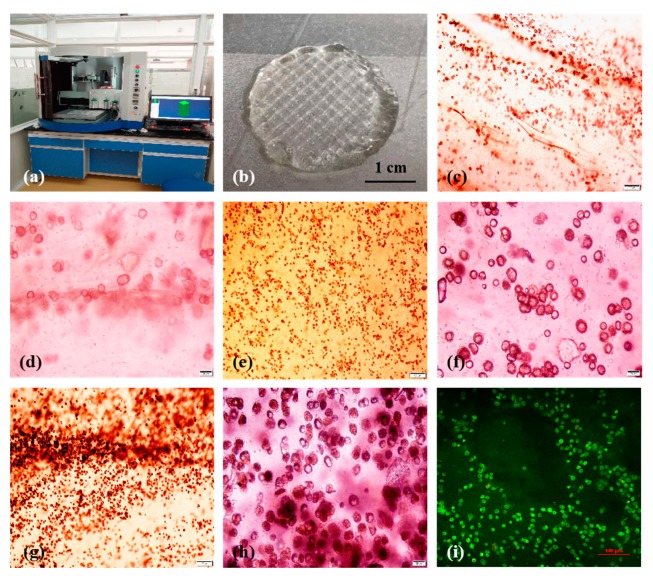
Three-dimensional (3D) bioprinting of cell-laden alginate/gelatin constructs for in vitro cultures: (**a**) a home-made three-nozzle 3D bioprinter, (**b**) a grid 3D construct made of SH-SY5Y cell-laden alginate/gelatin hydrogel, (**c**) an optical microscope image of SH-SY5Y cells in the 3D construct after 1 day of in vitro culture, (**d**) a magnified image of (**c**), (**e**) an optical microscope image of SH-SY5Y cells in the 3D construct after 3 days of in vitro culture, (**f**) a magnified image of (**e**), (**g**) an optical microscope image of SH-SY5Y cells in the 3D construct after 5 days of an in vitro culture, (**h**) a magnified image of (**g**), (**i**) an acridine orange/propidium iodide staining of the SH-SY5Y cells in the 3D construct after 5 days of in vitro culture.

**Figure 12 molecules-25-00756-f012:**
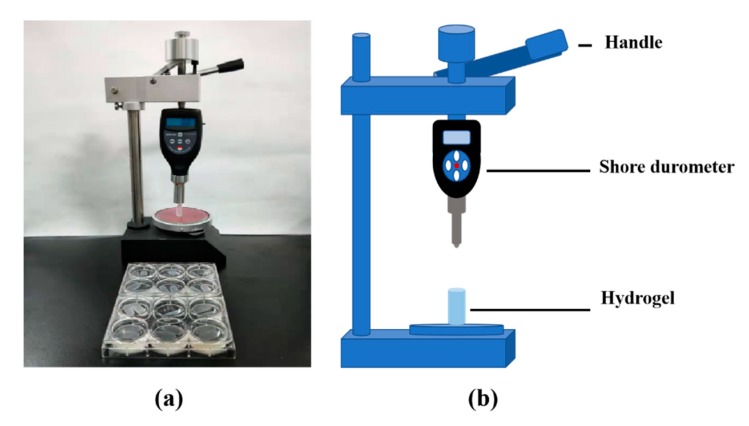
Schemical description of the hardness testing equipment (**a**) and measurement mechanism (**b**).

**Table 1 molecules-25-00756-t001:** Formulations of hydrogels.

Group	1	2	3	4	5	6	7	8	9	10	11	12
Gelatin (*w*/*v*%)	2.00	2.00	2.00	2.00	2.00	2.00	2.00	1.50	2.50	3.00	4.00	5.00
Alginate (*w*/*v*%)	0.50	0.75	1.00	1.25	1.50	2.00	2.50	0.75	0.75	0.75	0.75	0.75

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
