# Peer review of "An Interpenetrating Alginate/Gelatin Network for Three-Dimensional (3D) Cell Cultures and Organ Bioprinting"

_molecules, 2020, doi:10.3390/molecules25030756_

Round 1

Reviewer 1 Report

In this work Chen and coworkers studied alginate/gelatine IPNs, studying the effect of the activation of two distinct crosslinking procedures (ionic and enzymatic) on the properties of the idrogels, changing the relative amounts of the two component and the order of the crosslink (semi-IPN).

The work is well conducted, especially considering that the working group has a huge experience on similar systems(Ref 14-23). This affect the novelty of manuscript but, on the other hand, the scientific soundness is high.

I don't have knowledge on the biological part of the manuscript, so I won't comment that part.

I have only few comments:

on page 3 line 87 there is a typo (should be w/v and not w/W) in the same page, there is a citation reported that should not be there (line 102) the authors reported about the importance of the order of the two crosslinking mechanisms, could they  comment on the extent of crosslinking (i.e. amount of extractable monomers after the different stages of crosslnking)? could the authors compare some of the mechanical properties achieved from these hydrogels with the mechanical properties of ECM?

Author Response

Comments: In this work Chen and coworkers studied alginate/gelatine IPNs, studying the effect of the activation of two distinct crosslinking procedures (ionic and enzymatic) on the properties of the hydrogels, changing the relative amounts of the two component and the order of the crosslink (semi-IPN).

Response: Yes, the comments are all right.

Comments: The work is well conducted, especially considering that the working group has a huge experience on similar systems (Ref 14-23). This affect the novelty of manuscript but, on the other hand, the scientific soundness is high.

Response: Thanks a lot for the sincere comments!

Comments: I don't have knowledge on the biological part of the manuscript, so I won't comment that part.

Response: This is a very kind person with a heart of gold.

Comments: I have only few comments:

on page 3 line 87 there is a typo (should be w/v and not w/W) in the same page, there is a citation reported that should not be there (line 102). the authors reported about the importance of the order of the two crosslinking mechanisms, could they comment on the extent of crosslinking (i.e. amount of extractable monomers after the different stages of crosslnking)? could the authors compare some of the mechanical properties achieved from these hydrogels with the mechanical properties of ECM? 

Responses: The typo “w/w” has been corrected (i.e. w/v) and the wrong citation has been cancelled (Ref. 25).  

With regard to the extent of crosslinking, we have tried to ensure the complete reactions. Based on the reference 24 (i.e. Ref. 24), the amount of TG used was enough for fully reaction of gelatin molecules. Based on our previous experience (Ref. 14-23), we have chosen 2% (w/v) CaCl2 solution and increased the volume of 2% (w/v) CaCl2 solution to ensure the complete reaction of alginage molecules. After the double crosslinking, there are no extractable monomers during the later in vitro cell culture stages before the the breakages of the constructs.

It is regretful that there is no comparability between the mechanical properties of the mechanical properties achieved from these hydrogels with the mechanical properties of ECMs. On the one hand, we cannot measure the true hardness of human/animal ECMs without cells. On the other hand, the physiochemical propeties of different human/animal tissues, such as skin and adipose tissues, differ greatly. For example, the skin tissue of fetal rat 5 days after birth was significantly different from that 3 months after birth. When the cells are extracted from the living tissues, the sizes and shapes of the ECMs cannot be remained. So as to the hardnesses.

Reviewer 2 Report

The paper deals with the cell properties of interpenetrating alginate/gelatin network. The paper has high potential to be interesting, however it must be significantly improved.

Title of the paper. There is “bioprinting” but the paper is not about bioprinting. Bioprinting is not object of investigation. It is rather a method, how the samples were prepared. There is not any systematical investigation of bioprinting technique or parameters.

Introduction: It is unnecessary to explain the reader, what is polymer, monomer, repeating units. I would remove all the paragraph from line 39-45. As well the paragraphs about alginate gels (line 54 – 62) and Gelatin (line 65-71) should be reduced. Instead that useless information, please describe in more in detail problematics of interpenetrating networks, their mechanical properties and their potential to the cell culture. Mainly focus to the Gelatin/Alginate network. The authors frequently cite the paper of Cai Wen,  Lingling Lu,  Xinsong Li; [Mechanically Robust Gelatin–Alginate IPN Hydrogels by a Combination of Enzymatic and Ionic Crosslinking Approaches; Macromol. Mater. Eng. 2014, 299, 504–513; https://doi.org/10.1002/mame.201300274].

In Material and methods, I would remove the Figure 1. It is better to present the schematic picture of hardness measurement, if you would like to propose some picture.

Experimental

As I understood the main focus of the paper is the cell proliferation in the IPN. It is information, which makes the paper potentially interesting for the readers. However, there is missing information about the A-semi-IPN and G-semi-IPN hydrogels. The authors state that the constructs are broken during the first several days of in vitro cultures. However they certainly investigated the cell proliferation of semi IPNs before the material destruction. The A-G-IPNs are analyzed after 1-3 days.  The same analysis could be performed also for semi IPNs’. It is essential information to understanding of all effects described in the article.

 I would like to see, whether the semi IPN materials were broken due to destruction by cells or the material was thermally degraded. The figures would enable one to discuss the reason why the combined IPN was significantly more resistant to the cells than partial semi IPNs. One can see also, whether the IPN has some synergic effect to the cell proliferation or the effect is more or less additive from these partial networks.   

The experimental description of mechanical properties could be there as an additive information. It is not so new. There were published many papers about mechanical properties of IPNs in recent years. There can be observed some synergic effects, however, similar effects were recently published in literature.

Author Response

Comments: The paper deals with the cell properties of interpenetrating alginate/gelatin network. The paper has high potential to be interesting, however it must be significantly improved.

Response: The paper has been significantly improved.

Comments: Title of the paper. There is “bioprinting” but the paper is not about bioprinting. Bioprinting is not object of investigation. It is rather a method, how the samples were prepared. There is not any systematical investigation of bioprinting technique or parameters.

Response: The “bioprinting” in the title of the paper has been changed to “organ bioprinting”. This section has been revised in details.

Comments: Introduction: It is unnecessary to explain the reader, what is polymer, monomer, repeating units. I would remove all the paragraph from line 39-45. As well the paragraphs about alginate gels (line 54 – 62) and Gelatin (line 65-71) should be reduced. Instead that useless information, please describe in more in detail problematics of interpenetrating networks, their mechanical properties and their potential to the cell culture. Mainly focus to the Gelatin/Alginate network. The authors frequently cite the paper of Cai Wen, Lingling Lu, Xinsong Li; [Mechanically Robust Gelatin–Alginate IPN Hydrogels by a Combination of Enzymatic and Ionic Crosslinking Approaches; Macromol. Mater. Eng. 2014, 299, 504–513; https://doi.org/10.1002/mame.201300274].

Responses: In Introduction, the original paragraph from line 39-45 has been deleted. The paragraphs about alginate gels (line 54 – 62) and Gelatin (line 65-71) have been reduced. More in detail problematics of interpenetrating networks, their mechanical properties and their potential to the cell culture, have been added (page 2, line 70-72). The recommended paper of Cai Wen, Lingling Lu, Xinsong Li; [Mechanically Robust Gelatin–Alginate IPN Hydrogels by a Combination of Enzymatic and Ionic Crosslinking Approaches; Macromol. Mater. Eng. 2014, 299, 504–513; https://doi.org/10.1002/mame.201300274]  has been cited frequently (Ref. 26).

Comments: In Material and methods, I would remove the Figure 1. It is better to present the schematic picture of hardness measurement, if you would like to propose some picture.

Responses: In Material and Methods, the original Figure 1 has been substituted by the  schematic picture of hardness measurement (Figure 2).

Comments: Experimental

As I understood the main focus of the paper is the cell proliferation in the IPN. It is information, which makes the paper potentially interesting for the readers. However, there is missing information about the A-semi-IPN and G-semi-IPN hydrogels. The authors state that the constructs are broken during the first several days of in vitro cultures. However they certainly investigated the cell proliferation of semi IPNs before the material destruction. The A-G-IPNs are analyzed after 1-3 days. The same analysis could be performed also for semi IPNs’. It is essential information to understanding of all effects described in the article.

Response: Unfortunately the G-semi-IPN hydrogels are too weak to be used for encapsulating the cells and in vitro cultures. The constructs break immediately when they are put into culture medium.

Comments: I would like to see, whether the semi IPN materials were broken due to destruction by cells or the material was thermally degraded. The figures would enable one to discuss the reason why the combined IPN was significantly more resistant to the cells than partial semi IPNs. One can see also, whether the IPN has some synergic effect to the cell proliferation or the effect is more or less additive from these partial networks.   

Responses: We speculate that the enzymatic G-semi-IPN hydrogels were broken due to the weak covalent bonds. The even micropore structure is benefit for the synergic effects on hardness achievement and cell cultures of the IPNs.   

Comments: The experimental description of mechanical properties could be there as an additive information. It is not so new. There were published many papers about mechanical properties of IPNs in recent years. There can be observed some synergic effects, however, similar effects were recently published in literature.

Responses: Though the mechanical properties are not new, they are useful here for the overall awareness of the alginate/gelatin interpenetrating networks.  

Reviewer 3 Report

The authors aimed to demonstrated that an interpenetrating network of alginate and gelatin with a double crosslink would lead to a hydrogel with more attractive features for 3D cell culture. My first concern is related to alginate source. Although authors state that is from sigma, no characterization was made regarding its molecular weight, viscosity, M/G ratio, endotoxin content. In my opinion this characterization is mandatory.

Other comments that should be addressed before consider this manuscript for publication:

1) section 2.2. The CaCl2concentration is always the some independent of alginate concentration. CaCl2 shouldn´t be adjust according to alginate concentration?

2) section 2.6. Explain the cell source. Hydrogel formulation is missing in this section (0.75% Alg /2% gelatin). Mechanical properties studies in the presence of cells and throughout time are missing. Is gelatin degraded by the cells?

3) section 2.7. Cell density (2x106cells/mL)for cell viability (CCK-8) was lower than in section 2.6 (2x107cells/mL). There is any explanation for that?

4) Are cells able to establish cell-matrix interactions? Confocal images at higher magnification are missing. Are cells able to secrete ECM proteins?

5) section2.9. Please explain the hydrogel formulation for 3D bioprinting (1.5 gelatin/4%alginate). This formulation was not previously characterized. Therefore its mechanical properties are missing. Moreover, double gelation with calcium was made, what are the implications on cell viability and overall mechanical properties.

Author Response

Comments: The authors aimed to demonstrated that an interpenetrating network of alginate and gelatin with a double crosslink would lead to a hydrogel with more attractive features for 3D cell culture. My first concern is related to alginate source. Although authors state that is from sigma, no characterization was made regarding its molecular weight, viscosity, M/G ratio, endotoxin content. In my opinion this characterization is mandatory.

Response: We have checked our orders, and asked the sigma company about the parameters according to the catalog number: W201502-1KG. However, till now we have not received the response.  

Comments: Other comments that should be addressed before consider this manuscript for publication:

1) section 2.2. The CaCl2 concentration is always the some independent of alginate concentration. CaCl2 shouldn´t be adjust according to alginate concentration?

Response: We have added the amounts of the 2% (w/v) CaCI2 solutions when the alginate concentrations are increased (page 3, line 94-96).

2) section 2.6. Explain the cell source. Hydrogel formulation is missing in this section (0.75% Alg /2% gelatin). Mechanical properties studies in the presence of cells and throughout time are missing. Is gelatin degraded by the cells?

Response: The cell source has been added. Hydrogel formulation, i.e. 0.75% (w/v) alginate/2% (w/v) gelatin, has been added. Mechanical properties studies on hydrogels loaded with cells have not carried out. We have divided our studies into 3 parts: physiochemical properties studies to find a proper concentration of alginate/gelatin combination for A-G-IPN formation, biological tests to explore the possibility for in vitro 3D cell cultures based on the selected A-G-IPN, and 3D bioprinting using the cell-laden A-G-IPN.

3) section 2.7. Cell density (2 x106cells/mL) for cell viability (CCK-8) was lower than in section 2.6 (2 x107cells/mL). There is any explanation for that?

Response: Yes, cell density for cell viability (CCK-8) in section 2.7 was lower than in section 2.6 (2 x107 cells/mL). This is mainly due to the cell-laden hydrogel preparation methods. In section 2.7, 1 uL of hydrosol with 2 x106 cells/ml was used for each well of a 96-well plate, while in section 2.6, 1 ml of hydrosol with 2 x107 cells/ml was used for each well of a 12-well plate. This make the cell viability (CCK-8) test clear enough without forming dead cell layer and breaking the 3D constructs.   

4) Are cells able to establish cell-matrix interactions? Confocal images at higher magnification are missing. Are cells able to secrete ECM proteins?

Response: We agree that more higher magnification images would be better for cell-matrix interactions. Nevertheless, at present it is difficult for us to accomplish this.

5) section 2.9. Please explain the hydrogel formulation for 3D bioprinting (1.5 gelatin/4%alginate). This formulation was not previously characterized. Therefore its mechanical properties are missing. Moreover, double gelation with calcium was made, what are the implications on cell viability and overall mechanical properties.

Response: Sorry for the negligence! The concentration of the alginate and gelatin solutions are the same as those in section 2.6 and 2.7. The concentration should be half of their original concentrations when the two polymer solutions are mixed together.

By the way, the other errors have been corrected in red color.

Reviewer 4 Report

This study presented an IPN hydrogel using alginate and gelatin to improve the physical properties of hydrogels for 3D printing. Below concerns should be addressed before further consideration.

In the Abstract, This phrases have to fixed. "using Ca2+ 19 to crosslink gelatin molecules and transglutaminase (TG) to 20 crosslink alginate molecules, " Analysis of physical properties of the IPN hydrogels were not enough. Elastic or rheological modulus should be provided IPN hydrogels generally show the elastic property. Cyclic mechanical loading test should be performed to show the elastic property of the IPN hydrogels (i.g., Highly elastic and tough interpenetrating polymer network-structured hybrid hydrogel for cyclic mechanical loading-enhanced tissue engineering, Chem Mater, 2017, 29, 8425-8432).

3. In bioprinting, the resolution of filaments and the fidelity of the 3D printed structure should be analyzed.

4. In statistical analysis, the ANOVA test should be used instead of the Student' t-test.

Author Response

Comments: This study presented an IPN hydrogel using alginate and gelatin to improve the physical properties of hydrogels for 3D printing. Below concerns should be addressed before further consideration.

Response: The comments are okay.

Comments: In the Abstract, This phrases have to fixed. "using Ca2+ 19 to crosslink gelatin molecules and transglutaminase (TG) to 20 crosslink alginate molecules, " Analysis of physical properties of the IPN hydrogels were not enough. Elastic or rheological modulus should be provided IPN hydrogels generally show the elastic property. Cyclic mechanical loading test should be performed to show the elastic property of the IPN hydrogels (i.g., Highly elastic and tough interpenetrating polymer network-structured hybrid hydrogel for cyclic mechanical loading-enhanced tissue engineering, Chem Mater, 2017, 29, 8425-8432).

Response: In the Abstract, this phrases have been fixed: "using Ca2+ to crosslink alginate molecules and transglutaminase (TG) to crosslink gelatin molecules" (page 1, line 19-20). We must admit that the analysis of physical properties is inadequate. We have carefully studied the recommended paper and found some similarities and differences in the experimental designs with ours. At present, we have no such experimental conditions. We have consulted the lowest quotation of the relevant instrument, which is about 200,000 RMB, far beyond what we can afford. We will pay attention to these physical properties in the future studies.

Comments: 3. In bioprinting, the resolution of filaments and the fidelity of the 3D printed structure should be analyzed.

Response: This is a primary study on organ 3D bioprinting. The resolution of filaments and the fidelity of the 3D printed structure have been analyzed (page 15, line 430-435).

Comments: 4. In statistical analysis, the ANOVA test should be used instead of the Student' t-test.

Response: We have changed the comparison method and revised the manuscript about statistical analysis: “The Student’s t-tests was used for comparison between A-semi-IPN and A-G-IPN groups under the same concentration in WHC test and hardness test. One-way ANOVA method was used for comparing CV”. The following figure shows the result of one-way ANOVA comparison by a GraphPad Prism 8.0 software. SPSS software showed the same result.

Round 2

Reviewer 3 Report

Comments: The authors aimed to demonstrated that an interpenetrating network of alginate and gelatin with a double crosslink would lead to a hydrogel with more attractive features for 3D cell culture. My first concern is related to alginate source. Although authors state that is from sigma, no characterization was made regarding its molecular weight, viscosity, M/G ratio, endotoxin content. In my opinion this characterization is mandatory.

Response: We have checked our orders, and asked the sigma company about the parameters according to the catalog number: W201502-1KG. However, till now we have not received the response.  

I understand that Sigma would not provide the information, but authors should try to do alginate characterisation. It would be very difficult to repeat these results without knowing molecular weight and G/M content. 

Author Response

Comments: I understand that Sigma would not provide the information, but authors should try to do alginate characterisation. It would be very difficult to repeat these results without knowing molecular weight and G/M content.

Response: The parameters of the alginate have been added (page 3, line 84). Fortunately, we have received the following report from the commercial agent.

Reviewer 4 Report

1. Analyses of mechanical properties through compression and/or tensile test are essential for hydrogel studies. For IPN hydrogels, elastic properties should be provided. 

2. Any results for the resolution of filaments haven't been shown in the page 15. Please refer other 3D printing papers how to measure the resolution of 3D printed filaments.

3. Authors changed the title to ".....Organ Bioprinting." However, there is no organ bioprinting data.

Author Response

Comments: 1. Analyses of mechanical properties through compression and/or tensile test are essential for hydrogel studies. For IPN hydrogels, elastic properties should be provided.

Response: Yes, analyses of mechanical properties through compression and/or tensile test are essential for hydrogel studies. At present, we can only provide the hardness values. It seems that the additional tests cannot be accomplished in a short period because we are in a special virus infection condition and do not know when the ban can be lifted.  

Comments: 2. Any results for the resolution of filaments haven't been shown in the page 15. Please refer other 3D printing papers how to measure the resolution of 3D printed filaments.

Response: One reason is as above. Another reason is that the resolution of filaments is not an important parameter in 3D organ bioprinting because it is realized that the size of most living cells is between 10-20 μm. 3D bioprinting makes sense only when the thickness of the printing filaments is larger than 10 μm. Within this thickness cells can adjust themselves according the the microenvironments (Page 15, line 440-445).  

Comments: 3. Authors changed the title to ".....Organ Bioprinting." However, there is no organ bioprinting data.

Response: We have planned to use the three-nozzle 3D bioprinter to print organs. The primary data in the present study will be certified shortly in the future systematic studies. Additionally, the expression of manuscript has been revised carefully in red color.   
